# Limitations of DNA Methylation Profiling in High-Grade Gliomas: Case Series [note 1]

**DOI:** 10.3390/diagnostics15243225

**Published:** 2025-12-17

**Authors:** Marcus N. Milani, Constance P. Chen, Lindsey Sloan, Elizabeth C. Neil, Aundeep Yekula, Garret Fitzpatrick, Liam Chen

**Affiliations:** 1Medical School, University of Minnesota, Minneapolis, MN 55455, USA; 2Department of Radiation Oncology, University of Minnesota, Minneapolis, MN 55455, USA; 3Department of Neurology, University of Minnesota, Minneapolis, MN 55455, USA; 4Department of Neurosurgery, University of Minnesota, Minneapolis, MN 55455, USA; 5Department of Laboratory Medicine and Pathology, University of Minnesota, Minneapolis, MN 55455, USA

**Keywords:** high grade gliomas, DNA methylation profiling, IDH-wildtype gliomas, molecular diagnostics, unclassifiable CNS tumors

## Abstract

**Background and Clinical Significance**: DNA methylation profiling has revolutionized the classification of central nervous system (CNS) tumors, providing insights into tumor prognosis, recurrence, and personalized treatments. However, a significant challenge remains in classifying rare or molecularly undefined high-grade gliomas (HGGs) that fail to match existing methylation reference classes. This study evaluates the clinical, histopathological, and molecular characteristics of three unclassifiable cases through a retrospective analysis. Methylation profiling was performed by the National Institute of Health based on the 11b6 and 12b6 of the Heidelberg classifier, as well as the National Cancer Institute/Bethesda classifier. The cases were evaluated for histopathological features, molecular markers, and clinical outcomes. **Case Presentation**: We present three adult patients with histologically confirmed HGGs that were unclassifiable by standard DNA methylation profiling. All patients presented with diverse clinical and radiographic findings. Histopathological examination confirmed high-grade glial neoplasms in each case. However, methylation profiling failed to yield clear matches for any known class. Instead, profiling suggested indeterminate IDH-wildtype neoplasms with aggressive clinical courses. Following treatment, one patient experienced disease progression and died, while the other two remained without evidence of recurrence at follow-up. **Conclusions**: These cases underscore the persistent diagnostic challenges posed by CNS tumors that are unclassifiable by current DNA methylation, highlighting the urgent need for expanded reference datasets. While methylation profiling has transformed the field of tumor diagnostics, its limitations still exist. Enhanced collaboration to broaden diagnostic categories is essential to broaden diagnostic classifiers. Until these tools are refined, integration of clinical, histological, and molecular findings is imperative to optimize patient management, improve classification accuracy, and optimize therapeutic outcomes. Unclassifiable HGGs represent a critical gap in CNS tumor diagnostics. Addressing this requires global collaboration to enrich methylation databases. In the interim, a multimodal diagnostic strategy remains essential for the management of these challenging tumors.

## 1. Introduction

DNA methylation profiling of CNS tumors has allowed for the further subclassification of tumors historically classified by histological and molecular features. A DNA methylation profile is generated by a random forest algorithm, developed by DKFZ and Heidelberg University, and consists of 82 CNS tumor methylation classes and 9 non-tumorous classes [1,2,3,4,5]. In many CNS tumors, like glioblastomas, the tool offers a more powerful predictor for recurrence and prognosis than traditional histopathology [6,7,8,9,10]. In the 2021 WHO Classification of CNS Tumors, DNA methylation profiling allowed for the identification of high-grade astrocytoma with piloid features and diffuse leptomeningeal glioneuronal tumor [11]. Additionally, anaplastic pilocytic astrocytoma with elevated mitosis and high-grade features were further classified based on alternations in MAPK, CDKN2A/B and ATRX genes. Patients in this new category faced worse outcomes compared to pilocytic astrocytomas but fared better than IDH wildtype glioblastoma [3]. Reflecting these broader changes, a case report by Morgacheva et al. demonstrates the practical and integral role of DNA methylation profiling in solving challenging clinical cases with unusual presentations allowing for more precise treatment [12].

Beyond diagnostic confirmation, DNA methylation profiling prompts revision of initial histologic diagnosis in 12–18% of cases of CNS tumors [12,13]. Such reassignments are particularly relevant in patients with histologically HGGs with lower-grade molecular profiles due to their improved median survival, highlighting the importance of a combined integrated diagnostic approach [14,15]. Furthermore, in glioblastoma different methylation subclasses have shown improved survival benefit after aggressive resection compared to other methylation groups [16]. However, methylation profiling is only as powerful as its reference set. A patient with a rare tumor type may not match with an existing methylation profile or match with a different subcategory but with a low calibrated score [17,18]. Utilizing the Heidelberg classifier, a threshold of 0.9 is used to match to all classification levels and scores below 0.3 are determined Not Elsewhere Classified (NEC) [19]. A large population study of 1124 CNS tumors demonstrated 15% of cases could not be assigned a DNA methylation class based on the 2022 12.5 algorithm [20]. Unclassifiable CNS tumors based on DNA methylation profiling were found in a younger population, recurrent tumors post radiotherapy, and lower tumor purity sample [3]. NEC tumors were associated with a longer time to treat compared to matched cases, and in a subpopulation of IDH wildtype glioblastomas based on histological diagnosis, they had a longer time to initiate treatment and lower survival [21]. These poor outcomes demonstrate the need for a collectivist approach among researchers to report and share novel DNA methylation profiling to expand the algorithms database to further improve diagnostic accuracy.

In this report, we highlight a case series of adults with high-grade gliomas unclassifiable by current DNA methylation profiling tools, underscoring the diagnostic limitations of existing classifiers and the need for expanded reference datasets to improve accuracy and clinical utility.

## 2. Case Presentation

### 2.1. Clinical Characteristics of the Patients

Three female patients were evaluated between October 2022 to May 2024, two of whom were over 70 years old. Two had a family history of cancer, and one had a personal history of cancer. Magnetic resonance imaging identified a solitary 4.2 cm solid and cystic mass in the right frontal lobe in Case 1, a hyperintense lesion involving the left thalamus in Case 2, and patchy abnormal contrast enhancement in the anterior left temporal lobe, left insula, and left lentiform nuclei in Case 3. Histopathological analysis confirmed HGGs in all cases, but further molecular characterization with methylation testing was not clarifying Two patients (Cases 1 and 3) received postoperative chemotherapy, while one (Case 2) opted for hospice care. At the time of writing, one patient (Case 2) is deceased, and two (Cases 1 and 3) remain without evidence of disease after one year of imaging surveillance.

### 2.2. Case Presentation

#### 2.2.1. Case 1

A 43-year-old woman with a history of latent tuberculosis and a family history of pancreatic cancer (maternal grandmother) presented with headaches, dizziness, and generalized weakness. MRI of the brain revealed a solitary 4.2 cm solid and cystic mass in the right frontal lobe. The mass effect, along with surrounding edema, caused obstruction at the foramen of Monro, leading to left ventricular entrapment and transependymal flow of CSF (Figure 1A). Right frontal craniotomy for resection of the mass was performed. Post-operative imaging showed mass debulking, with residual enhancing soft tissue at the surgical site measuring 2.8 cm, suspicious for residual tumor. One month later, she underwent a second resection followed by laser interstitial thermal therapy (LITT). A CT one week later showed post-procedural changes from the right frontal lobe mass laser ablation, with stable-appearing necrosis and hemorrhage along the medial aspects of the ablation zone. Three months after surgery, she completed upfront chemoradiotherapy with concurrent temozolomide. She then began adjuvant-dosed temozolomide and completed the planned six cycles of therapy. Follow up imaging following therapy demonstrated a subtle increase in T2 FLAIR signal around the right frontal lobe, likely representing post-radiation changes. Imaging three months later showed stability from a cancer standpoint but suggested possible radiation-induced optic neuritis. One year later, repeat imaging confirmed continued cancer stability but revealed notable worsening of expansile contrast enhancement in the right prechiasmatic optic nerve, likely due to radiation effects.

#### 2.2.2. Case 2

An 88-year-old female with a history of chronic episodic migraines with visual aura presented to the ED with right leg weakness, several months of forgetfulness, and recurrent visual auras. A CT of the head and neck revealed a notable mass in the left thalamus with mixed density, hypodense and isodense components, and localized mass effect effacing the third ventricle. A subsequent MRI demonstrated T2/FLAIR hyperintense signal involving the left thalamus with mild to moderate adjacent vasogenic edema and mass effect on the posterior aspect of the third ventricle and proximal cerebral aqueduct, along with mild hydrocephalus (Figure 1B). Because of the unresectable location of the tumor, she underwent biopsy of the lesion followed by LITT. Temozolomide was not recommended due to the patient’s performance status in the absence of MGMT promoter methylation. Ultrasound revealed extensive deep vein thrombosis with a nonocclusive clot in the common femoral and deep femoral veins extending into the external iliac vein. The patient was started on dexamethasone 2 mg daily but declined radiation therapy and transitioned to hospice care.

#### 2.2.3. Case 3

A 73-year-old female with Lynch syndrome due to a MSH2 mutation and a personal remote history of prior uterine cancer that was successfully treated, along with an extensive family history of cancer, presented to the ED for evaluation of aphasia and progressive fatigue. MRI brain imaging demonstrated abnormal T2 signal hyperintensity with patchy abnormal contrast enhancement involving the anterior third of the left temporal lobe, insula, and lentiform nuclei (Figure 1C). She underwent an awake left temporal craniotomy for mass resection. Post-operative MRI demonstrated postsurgical changes in the partially resected left temporal mass, including expansile T2 signal in the left basal ganglia, thalamus, midbrain, insula, front and temporal lobes with rightward midline shift. The patient then underwent a 6-week course of radiotherapy, followed by six cycles of temozolomide 320 mg. The patient had multiple contrast enhancing areas within the prior radiation field on follow up imaging for which she underwent a diagnostic re-resection. Pathology was consistent with radiation necrosis. She was started on bevacizumab and is currently without evidence of disease on surveillance imaging.

### 2.3. Histopathological Characteristics of the Tumors

These three cases of HGGs, not otherwise classified, demonstrate diverse histopathological and immunohistochemical features (Figure 2, Figure 3 and Figure 4). All cases retained ATRX expression, exhibited p53 overexpression, and had elevated Ki-67 indices, while being negative for IDH1. Case 1 shows a glial neoplasm with epithelioid, spindle, and gemistocytic morphology, featuring prominent perivascular inflammatory infiltrates, infiltrative growth, and neuronal differentiation. Immunohistochemically, it is GFAP-positive with weak synaptophysin expression, retains ATRX, and overexpresses p53, but is negative for Olig2 and BRAF V600E, with an elevated Ki-67 index. Case 2 is characterized by bizarre multinucleated giant cells and perivascular lymphocyte accumulation, without necrosis or microvascular proliferation. It demonstrates GFAP, OLIG2, and p53 positivity with retained ATRX and H3K27me3 expression, while being negative for IDH1, BRAF V600E, and H3K27M, and shows a high Ki-67 index. Case 3 consists of an infiltrating glioma with increased mitotic activity but absent microvascular proliferation or necrosis. Its immunoprofile reveals OLIG2, ATRX, and p53 positivity with retained MLH1 and PMS2, but is negative for IDH1, MSH2, and MSH6, along with a moderately elevated Ki-67 index (Table 1).

### 2.4. Molecular Characteristics of the Tumors

Mutation analysis revealed no IDH1 or IDH2 mutations, and no gene fusions were detected in any case. TP53 mutations were present in Cases 2 and 3, while MGMT promoter methylation was positive only in Cases 1 and 3. Case 1 demonstrated unknown mutations in PDGFRA, GNA11, and PPM1D, but none in H3-3A, BRAF, or the TERT promoter. Positive for MGMT methylation. It displayed a near-haploid/pseudohyperdiploid genome with chromosome 7 gain and copy-neutral loss of heterozygosity on chromosome 10, without CDKN2A/B homozygous deletion. Case 2 exhibited complex chromosomal copy number alterations including chromothripsis of chromosome 2, losses of 7p22.3p21.3, 9p, 11p, chromosome 12, 13q11q12.11, 13q14.11q14.3 (encompassing RB1), 13q21.31q21.32, 14q12q32.33, 17p13.3p11.2 (including TP53), multiple 17q gains, and 19q13.4 loss. Case 3 harbored TP53 mutations (R273H and R175H), a PDGFRA mutation, and MGMT methylation positivity. It showed losses of chromosomes 2 and 5, 9p24.3p21.1 (including CDKN2A/B), and chromosome 17 gain (Table 2).

### 2.5. DNA Methylation Profiling of the Tumors

In all three cases, DNA methylation-based tumor classification using versions 11b6 and 12b6 of the Heidelberg classifier, as well as the National Cancer Institute/Bethesda classifier, did not yield a definitive match to any established methylation class. Specimens were collected in fresh and formalin-fixed samples consisting of fragments, multiple chunks, and tumor fluid. All tumor samples were adequate for analysis, with estimated fraction of viable lesioned cells at 70%, 60% and 60% for Cases 1, 2, and 3, respectively. The immunoperoxidase and in situ hybridization tests performed for diagnosis were developed by the Laboratory of Pathology, NCI. The methylation profiles in each case were consistent with a high-grade neoplasm. Cases 1, 2, and 3 all represent high-grade, IDH-wildtype glial neoplasms, which are associated with aggressive clinical behavior (Figure 5).

## 3. Discussion

All cases in this series were HGGs that could not be classified by the current standard DNA methylation profiling tools, specifically the Heidelberg and National Cancer Institute classifiers. This diagnostic challenge underscores the existence of rare or as-yet undefined glioma subtypes that are not represented in current reference databases, a limitation acknowledged in the recent literature [20,21,22,23]. The inability to classify these tumors is clinically significant, as unclassifiable cases are associated with delayed treatment decisions, clinical uncertainty, and potentially worse outcomes, particularly in aggressive entities such as HGGs [23]. In clinical practice, a NEC designation may prolong the interval between biopsy and therapeutic planning, which is particularly detrimental for patients with rapidly progressive subsets of tumors.

The current generation of methylation classifiers may, in some instances, broaden rather than refine diagnostic categories, especially when reference sets are incomplete or when classifier thresholds are not optimized for rare entities [21,22,24]. For instance, Case 1 demonstrated a gain of chromosome 7 and loss of chromosome 10 with a CDKN2A/B homozygous deletion. Similar chromosomal microarray patterns have been reported in giant cell glioblastomas and anaplastic pleomorphic xanthoastrocytomas [25,26]. However, the overall molecular profile of this case does not fully align with either entity. DNA methylation profiling placed the tumor in proximity to the High-Grade Astrocytoma with Piloid Features class, yet it did not achieve a definitive match. Case 1 also lacked typical molecular features of this entity, such as alterations in CDKN2A/B, NF1, BRAF, FGFR1, and ATRX [27]. These results suggest this case may represent a new unique entity sharing characteristics with several types of high-grade gliomas. While a neuropathologist might have considered several possible diagnoses based on its canonical features, the indeterminate methylation results ultimately created diagnostic ambiguity rather than clarity.

Nevertheless, the detection of unusual mutations, such as alterations in platelet-derived growth factor receptor alpha, suggests that some of these unclassifiable tumors may harbor actionable molecular targets. These findings emphasize that a NEC result should not conclude the end of diagnostic inquiry; instead, it should prompt additional comprehensive molecular profiling, including next-generation sequencing and transcriptomic analysis, to identify actionable targets or refine subclassification. This integrated approach not only strengthens diagnostic confidence but may also guide enrollment in precision oncology trials and inform individualized treatment strategies [20,21,28].

Although the unclassifiable nature of these cases contributed to diagnostic uncertainty, its direct impact on survival varied. In Case 2, for instance, patient-specific factors led to the shared decision against standard-of-care treatment. While age and quality of life ultimately determine the aggressiveness of therapy, diagnostic ambiguity can complicate medical judgment and decision-making for both clinicians and patients. Further research is needed to understand how such uncertainty influences therapeutic choices and whether it leads to a self-fulfilling prophecy of minimizing treatment due to the perceived uncertainty of the disease.

In Case 2, the DNA methylation profiling was conclusively negative. Its proximity to the control reaction group, along with perivascular accumulation of lymphocytes, suggestive of inflammation, highlights important considerations regarding tumor heterogeneity and sample quality. A NEC designation is formally assigned when the classifier scores fall below 0.3, an outcome influenced by intratumoral heterogeneity, low tumor cell content, or poor sample integrity [3]. Given the large viable tumor sample assessed by the NCI in Case 2, this likely represents a true NEC case. In such cases, interpreting DNA methylation within the histopathological context remains essential.

Case 3 has a history of Lynch syndrome, confirmed by loss of MSH2 and MSH6 expression. The association between Lynch syndrome and primary brain tumors is well-established, with a reported lifetime risk of brain tumors ranging from 1% to 6% [29,30,31]. This risk is particularly elevated in patients, like Case 3, with MSMH2 gene mutations [32]. Immunohistochemical analysis shows the tumor cells have retained expression of MLH1 and PMS2 but demonstrate a complete loss of MSH2 and MSH6. These results are consistent with the tumor’s high mutation burden and the patient’s known MSH2 germline mutation, indicating a second, somatic MSH2 mutation. However, the relationship between the patient’s Lynch syndrome and the tumor’s lack of methylation profiling remains unclear. More research is needed on using DNA methylation to classify gliomas in patients with Lynch syndrome.

Integrating DNA methylation profiling into the diagnostic workflow for challenging CNS tumors has been shown to improve the definition of rare entities and to facilitate the discovery of novel tumor types [3,20,22]. Notably, methylation-based reclassification has already led to the recognition of new categories in the WHO 2021 framework [13], such as high-grade astrocytoma with piloid features and diffuse glioneuronal leptomeningeal tumors, underscoring its power to reshape classification paradigms. The recent update of the DKFZ classifier from version 11b4 to 12.8 resulted in the reclassification of 62.3% of previously NEC cases, as reported by Drexler et al. 2024, demonstrating the rapid evolution and increasing sensitivity of these tools [23]. Yet, even with iterative updates, between 8 and 15% of tumors remain unclassifiable, suggesting that current algorithms cannot fully capture the biological and epigenetic diversity of CNS tumors. This emphasizes the need for ongoing expansion of reference datasets to capture the full spectrum of glioma biology [3,20,22,23]. This will require coordinated global efforts to collect high-quality, rare tumor methylation profiles, as well as integration of pediatric, recurrent, and post-treatment samples that are currently underrepresented. In parallel, algorithmic improvements, and machine learning models trained on multi-institutional datasets should be explored as strategies to reduce NEC calls [33,34,35,36,37].

Ultimately, despite the shared diagnosis of high-grade glioma NEC, each patient’s case proved uniquely challenging due to highly variable molecular characteristics, histopathology, and clinical context. These tumors do not represent a unified entity and likely constitute rare subclassifications. Each case reveals both the strengths and the limitations of DNA methylation profiling.

## 4. Conclusions

In summary, these findings reinforce the value of DNA methylation profiling in CNS tumor diagnostics, while also illustrating its current clinical limitations. The NEC designation presents a multifaceted challenge: it can introduce diagnostic delays in treatment, and it can contribute to prognostic uncertainty that complicates shared decision-making. Continued refinement of classifiers and incorporation of additional molecular data are essential for improving diagnostic precision, guiding targeted therapy, and advancing the classification of HGGs [3,20,21,22,23]. Until such refinements are achieved, clinicians should interpret NEC calls with caution and employ an integrated approach that synthesizes histopathology, radiographic findings, and broad molecular testing to ensure optimal patient care.

## Figures and Tables

**Figure 1 diagnostics-15-03225-f001:**
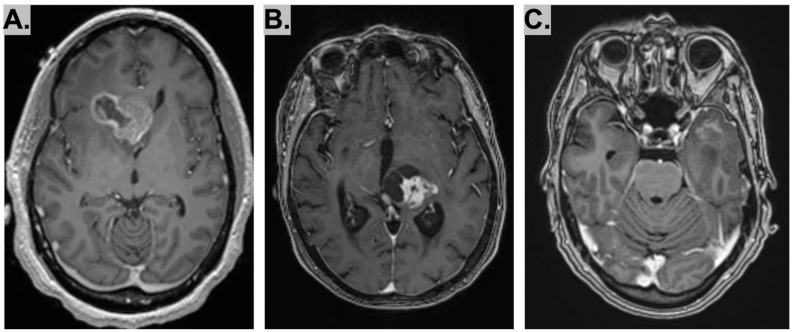
Representative image of (**A**–**C**) axial brain MRI for Cases 1–3. (**A**) The T1 3D MPRAGE Post-Contrast image demonstrates a right solid and cystic frontal mass. (**B**) The T1 3D MPRAGE Post-Contrast image demonstrates a heterogeneously enhancing cystic and solid mass lesion centered in the region of the left thalamus. (**C**) The T1 3D MPRAGE Post-Contrast demonstrates a T2 signal hyperintensity with patchy abnormal contrast enhancement involving the anterior third of the left temporal lobe.

**Figure 2 diagnostics-15-03225-f002:**
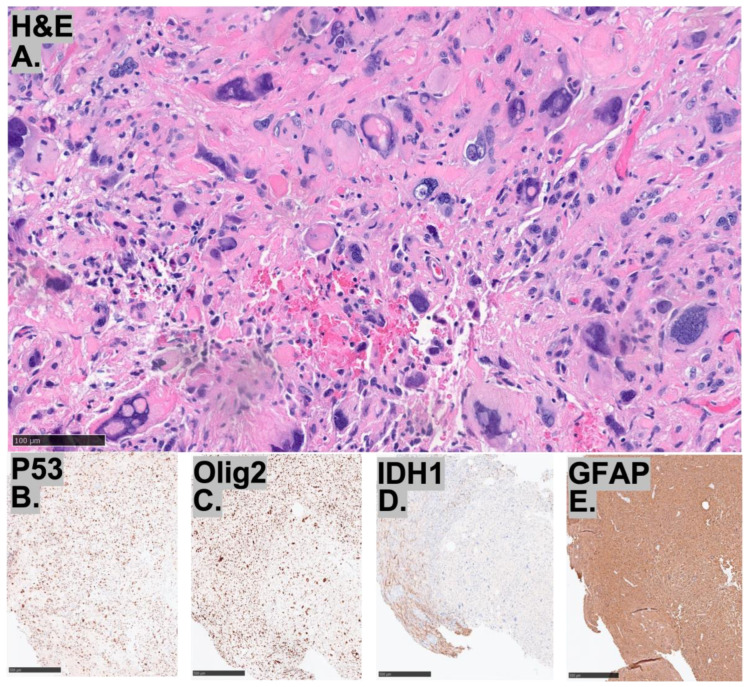
Representative image of (**A**–**E**) histopathological findings for Case 1. (**A**) The H&E shows an infiltrative growth pattern and brisk mitotic activity, bar = 100 μm.. Immunohistochemistry demonstrates patchy overexpression of P53 (**B**), lack of Olig2 (**C**), IDH-1 (**D**), and strong expression for GFAP (**E**) bar = 500 μm.

**Figure 3 diagnostics-15-03225-f003:**
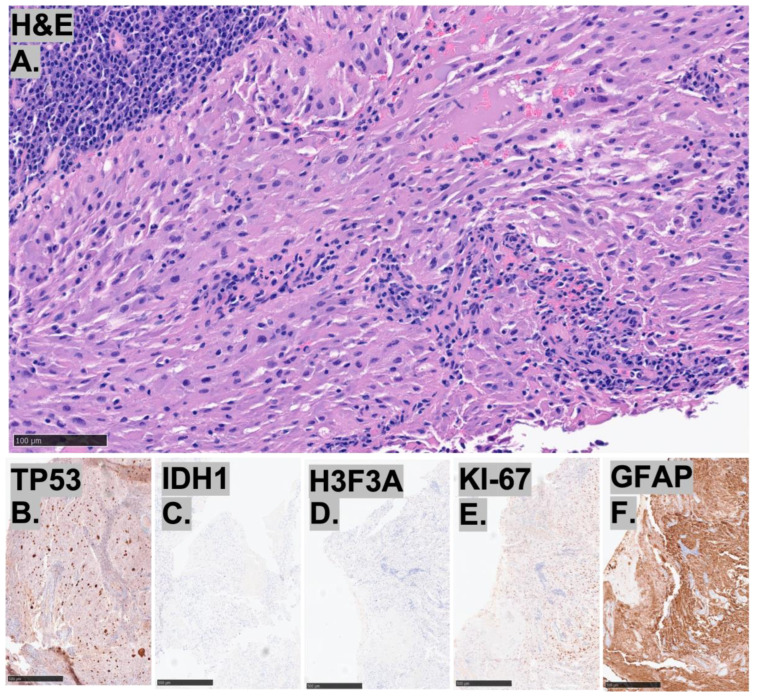
Representative images of (**A**–**F**) histopathological findings for Case 2. The H&E (**A**) shows large, bizarre multinucleated giant cells and perivascular accumulation of lymphocytes. Immunohistochemistry demonstrates strong expression for p53, bar = 100 μm (**B**), negative for IDH1 (**C**), H3F3A (**D**), and positive for GFAP (**F**). Ki-67 index estimated to be 20% (**E**) bar = 500 μm.

**Figure 4 diagnostics-15-03225-f004:**
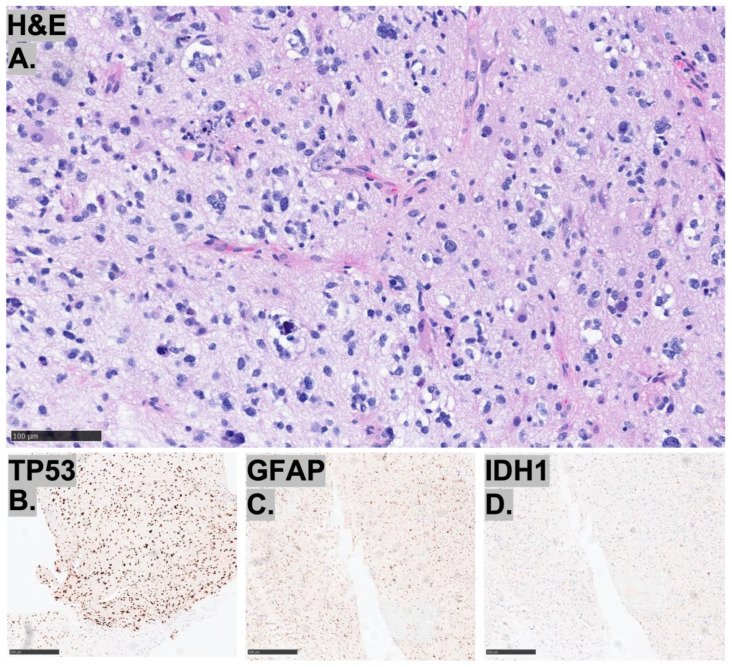
Representative images of (**A**–**D**) histopathological findings for Case 3. (**A**) The H&E shows infiltrating glioma with increased mitotic activity, but no microvascular proliferation or necrosis, bar = 100 μm. Immunohistochemistry was positive for p53 (**B**), GFAP (**C**), and negative for IDH1 (**D**) bar = 500 μm.

**Figure 5 diagnostics-15-03225-f005:**
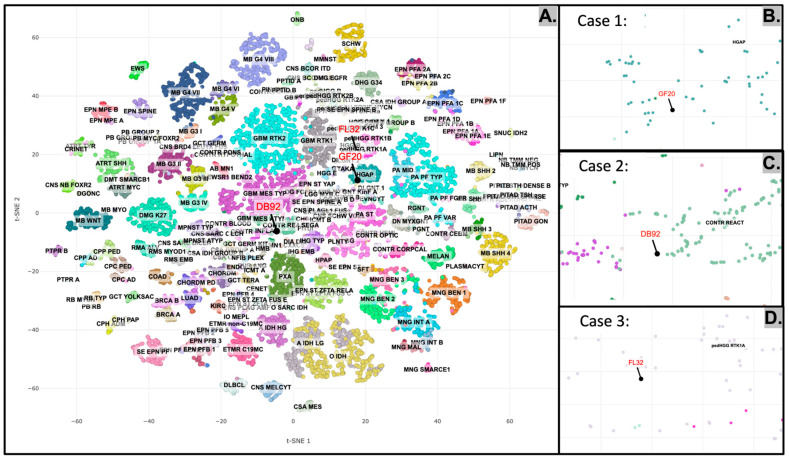
Representative images (**A**–**D**) of tSNE projections of tumor methylation classes from published brain malignancy methylation datasets. Cases 1–3 (**B**–**D**) show no match to known methylation classes.

**Table 1 diagnostics-15-03225-t001:** The immunohistochemical profiles of the three cases. Positive (+) indicates positive findings. Negative (−) indicates negative findings. Empty fields indicate stains not performed.

Histopathology	Case 1	Case 2	Case 3
ATRX	+	+	+
p53	overexpression	overexpression	overexpression
Ki-67 index	elevated	elevated	elevated
IDH1	−	−	−
GFAP	+	+	+
Synaptophysin	weakly positive		
Olig2	−	patchy positivity	+
BRAF V600E	−	−	
H3K27M		−	

**Table 2 diagnostics-15-03225-t002:** The molecular profiles of the three cases. Positive (+) indicates positive findings. Negative (−) indicates negative findings. Empty fields indicate tests not performed.

Molecular	Case 1	Case 2	Case 3
IDH1/2	−	−	−
TP53	Normal	Pathologic mutation	Pathologic mutation
ATRX	−	−	−
BRAF	−	−	−
CDKN2A	−	−	−
CTNNB1		−	−
EGFR	−	−	−
FGFR1/2/3	−	−	−
HF3A	−	−	−
HIST1H3B	−	−	−
MET		−	−
PDGFRA	Variant	−	Pathologic mutation
PTEN		−	−
PTPN11		−	−
TERT	−	−	−
GNA11	Variant		
PPM1D	Variant		
MGMT promoter methylation	+	−	+
Gene fusion events	−	−	−
Chromosomal microarray	Near-haploid/pseudohyperdiploid genome with gain of chromosome 7 and copy neutral loss of heterozygosity of chromosome 10.	Chromosome copy number complexity with chromothripsis of chromosome 2, loss of 7p22.3p21.3, loss of 9p, loss of 11p, loss of chromosome 12, loss of 13q11q12.11, loss of 13q14.11q14.3 (including RB1), loss of 13q21.31q21.32, loss of 14q12q32.33, loss of 17p13.3p11.2 (including TP53), multiple level gain of 17q, and loss of 19q13.43.	Loss of chromosomes 2 and 5, loss of 9p24.3p21.1 (including CDKN2A and CDKN2B), and gain of chromosome 17.

## Data Availability

The original contributions presented in this study are included in the article. Further inquiries can be directed to the corresponding author.

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
