# Peer review of "Limitations of DNA Methylation Profiling in High-Grade Gliomas: Case Seriesâ€"

_diagnostics, 2025, doi:10.3390/diagnostics15243225_

Round 1
Reviewer 1 Report
Comments and Suggestions for Authors
This study focuses on cases of high-grade gliomas that cannot be classified by DNA methylation analysis, which is a clinically challenging and under researched field. The research results have potential significance for improving the diagnosis and treatment of central nervous system tumors, but the following key issues need to be addressed to enhance the attractiveness of the manuscript to readers of Diagnostics.
Major remarks:
1. The patient in Case 2 did not receive standard treatment due to their advanced age and performance status, and their adverse outcomes are more likely to be related to inadequate treatment rather than solely due to the "unclassifiable" nature of the tumor, which to some extent weakens the argument for directly linking outcomes with limitations in diagnostic techniques.
2. The discussion section extensively restates the introduction and existing literature, but lacks close integration with the findings of this study itself. The similarities and differences among the three cases should be compared, and their commonalities should be explored in depth to provide insights for methylation classification.
Minor remarks:
1.The contrast of MRI images in the three case figures is poor. It is recommended to readjust the contrast of MRI images to achieve the best lesion display effect.
Author Response
Comment 1: The patient in Case 2 did not receive standard treatment due to their advanced age and performance status, and their adverse outcomes are more likely to be related to inadequate treatment rather than solely due to the "unclassifiable" nature of the tumor, which to some extent weakens the argument for directly linking outcomes with limitations in diagnostic techniques.
Response 1: We agree that the adverse outcome in Case 2 cannot be attributed solely to the unclassifiable nature of the tumor. In response, we have added a paragraph to broaden the discussion on the impact of NEC on clinical decisions, while acknowledging this limitation. Furthermore, we have expanded our analysis of Case 2 to emphasize how diagnostic ambiguity itself can complicate an already challenging clinical scenario, regardless of other contributing factors.
Lines: Lines 269-284
Comment 2: The discussion section extensively restates the introduction and existing literature, but lacks close integration with the findings of this study itself. The similarities and differences among the three cases should be compared, and their commonalities should be explored in depth to provide insights for methylation classification.
Response 2: We have expanded the discussion to analyze the three cases in greater detail. Using each case as a springboard, we examine the differences to explore the challenges in working up NEC HGGs. We hope this deeper integration highlights our institution’s experience and draws more specific insight for methylation classification.
Lines: 250-260; 269-295; 313-317
Comment 3: The contrast of MRI images in the three case figures is poor. It is recommended to readjust the contrast of MRI images to achieve the best lesion display effect.
Response 3: Reselected the images for better clarity and selected T1 post contrast sequence.
Lines: Figure 1
Reviewer 2 Report
Comments and Suggestions for Authors
This manuscript presents a case series investigating the limitations of DNA methylation profiling in classifying three rare or poorly understood high-grade gliomas (HGGs). The authors retrospectively analyze clinical, histopathological, and molecular data from three adult patients whose tumors could not be matched to a known class using standard methylation classifiers (Heidelberg and NCI/Bethesda). All cases were confirmed as HGGs by MRI and histopathology, yet methylation profiling yielded indeterminate results for IDH-wildtype neoplasms. The study highlights a significant diagnostic challenge in neuro-oncology and underscores the necessity of integrating multimodal data for patient management while advocating for expanded reference databases to improve the diagnostic power of methylation profiling.
The authors address a clinically relevant and timely topic regarding the limitations of DNA methylation profiling in diagnosing challenging high-grade glioma cases. The case series format is appropriate for illustrating this diagnostic dilemma. However, several critical aspects require clarification and further elaboration to strengthen the manuscript's validity and impact. The findings, while interesting, would benefit from a more robust presentation and a deeper discussion of the implications of the methylation results.
- Patient Outcomes and Follow-up:The clinical follow-up for the two surviving patients (Cases 1 and 3) requires more precise definition. Given that the cases span 2022-2024, the current statement of "no evidence of recurrence" is difficult to interpret without specific follow-up durations. Please provide the exact length of follow-up for each patient from diagnosis/surgery to the last clinical or radiological assessment. This is crucial for contextualizing the "aggressive clinical courses" mentioned and for assessing the true prognostic significance of an "unclassified" methylation status.
- Section 3.3 Histopathological characteristics:The descriptive text, while supported by Figures 1-3, would be significantly enhanced by a summary table. A consolidated table presenting key histopathological features for each case would allow for direct comparison and improve readability.
- Section 3.4 Molecular characteristics:Similarly, the molecular findings are critically important yet are not synthesized for easy review. A table summarizing the results for all three cases is strongly recommended.
- In 3.5 DNA methylation profiling of the tumors (Figure 4):
Clarity of Figure 4A: The resolution of Figure 4, Panel A, is currently insufficient to discern the critical details. The figure must be improved to allow the reader to verify the authors' interpretation.
Classification Ambiguity: There is an apparent contradiction that needs to be addressed. The manuscript notes that Case 2 appears to cluster near the HGAP group in the t-SNE plot (Figure 4A), yet it is reported as unclassified. The authors must provide a detailed explanation in the results or discussion for this discrepancy. What was the calibrated score or similarity threshold for this case? Does its proximity to HGAP suggest a potential biological relationship, and if so, why was a definitive classification not possible?
Representativeness of Methylation Data: The t-SNE plot in Figure 4 uses a single data point to represent the methylation profile of each entire tumor. The authors should briefly comment in the methods or figure legend on the technical approach (e.g., was this from a single biopsy region or an averaged profile from multiple tumor areas?) and acknowledge the potential impact of intra-tumoral heterogeneity on the classification outcome.
Author Response
Comment 1: The authors address a clinically relevant and timely topic regarding the limitations of DNA methylation profiling in diagnosing challenging high-grade glioma cases. The case series format is appropriate for illustrating this diagnostic dilemma. However, several critical aspects require clarification and further elaboration to strengthen the manuscript's validity and impact. The findings, while interesting, would benefit from a more robust presentation and a deeper discussion of the implications of the methylation results.
Response 1: We added more detailed discussion on the implication of the methylation results in our individual cases.
Lines: 250-260; 269-295; 313-317
Comment 2: Patient Outcomes and Follow-up:The clinical follow-up for the two surviving patients (Cases 1 and 3) requires more precise definition. Given that the cases span 2022-2024, the current statement of "no evidence of recurrence" is difficult to interpret without specific follow-up durations. Please provide the exact length of follow-up for each patient from diagnosis/surgery to the last clinical or radiological assessment. This is crucial for contextualizing the "aggressive clinical courses" mentioned and for assessing the true prognostic significance of an "unclassified" methylation status.
Response 2: Specified that patients remained disease free at one year imaging surveillance.
Lines: 100
Comment 3: Section 3.3 Histopathological characteristics:The descriptive text, while supported by Figures 1-3, would be significantly enhanced by a summary table. A consolidated table presenting key histopathological features for each case would allow for direct comparison and improve readability. Section 3.4 Molecular characteristics:Similarly, the molecular findings are critically important yet are not synthesized for easy review. A table summarizing the results for all three cases is strongly recommended. In 3.5 DNA methylation profiling of the tumors (Figure 4): Clarity of Figure 4A: The resolution of Figure 4, Panel A, is currently insufficient to discern the critical details. The figure must be improved to allow the reader to verify the authors' interpretation.
Response 3: We created tables to better present the histology and molecular data. We recreated Figure 4 for additional clarity.
Lines: Table 1 & 2; Figure 5
Comment 4: Classification Ambiguity: There is an apparent contradiction that needs to be addressed. The manuscript notes that Case 2 appears to cluster near the HGAP group in the t-SNE plot (Figure 4A), yet it is reported as unclassified. The authors must provide a detailed explanation in the results or discussion for this discrepancy. What was the calibrated score or similarity threshold for this case? Does its proximity to HGAP suggest a potential biological relationship, and if so, why was a definitive classification not possible?
Response 4: We have addressed this by adding an explanation that details the calibrated score for Case 2. The text now discusses the biological suggestion of Case 1’s proximity to HGAP, as well as Case 2’s proximity to the control reaction. We have also further elaborated on the NEC classification score's interpretation.
Lines: 250-260; 277-284
Comment 5: Representativeness of Methylation Data: The t-SNE plot in Figure 4 uses a single data point to represent the methylation profile of each entire tumor. The authors should briefly comment in the methods or figure legend on the technical approach (e.g., was this from a single biopsy region or an averaged profile from multiple tumor areas?) and acknowledge the potential impact of intra-tumoral heterogeneity on the classification outcome.
Response 5: We have updated the DNA methylation methods section to detail the technical approach for sample selection. In the discussion, we also address the potential impact of intratumoral heterogeneity on the classification results.
Lines: 199-204; 280-284
Round 2
Reviewer 2 Report
Comments and Suggestions for Authors
There are no outstanding issues.